# Behind closed doors: Protective social behavior during the COVID-19 pandemic

Kyla Thomas[1]*, Peter G. Szilagyi[2], Sitaram Vangala[3], Rebecca N. Dudovitz[2], Megha D. Shah[4], Nathalie Vizueta[2], Arie Kapteyn[1]

1 Center for Economic and Social Research, Dornsife College of Letters Arts and Sciences, University of Southern California, Los Angeles, CA, United States of America, 2 Department of Pediatrics, David Geffen School of Medicine, University of California at Los Angeles, Los Angeles, CA, United States of America, 3 Department of Medicine Statistics Core, David Geffen School of Medicine, University of California at Los Angeles, Los Angeles, CA, United States of America, 4 Los Angeles County Department of Public Health, Office of Health Assessment and Epidemiology, Los Angeles, CA, United States of America

* kylathom@usc.edu

**Data Availability Statement:** All Understanding America Study (UAS) survey data files are available on the Understanding America Study website

## Abstract

The success of personal non-pharmaceutical interventions as a public health strategy requires a high level of compliance from individuals in private social settings. Strategies to increase compliance in these hard-to-reach settings depend upon a comprehensive understanding of the patterns and predictors of protective social behavior. Social cognitive models of protective behavior emphasize the contribution of individual-level factors while social-ecological models emphasize the contribution of environmental factors. This study draws on 28 waves of survey data from the Understanding Coronavirus in America survey to measure patterns of adherence to two protective social behaviors–private social-distancing behavior and private masking behavior–during the COVID-19 pandemic and to assess the role individual and environmental factors play in predicting adherence. Results show that patterns of adherence fall into three categories marked by high, moderate, and low levels of adherence, with just under half of respondents exhibiting a high level of adherence. Health beliefs emerge as the single strongest predictor of adherence. All other environmental and individual-level predictors have relatively poor predictive power or primarily indirect effects.

## Introduction

Mitigating risk for future pandemics will depend upon the immediate and widespread adoption of personal non-pharmaceutical interventions (NPIs) including social-distancing and masking. The challenge of personal NPIs as a public health strategy is that they require sustained behavioral compliance on the part of individuals. Lockdowns and mask mandates ensure some compliance in public settings but restrictions are not always accepted or enforced, especially in private social settings where risk of infection is high [1] but mechanisms of enforcement are largely absent [2]. Anticipating infection risk and designing effective interventions to increase behavioral compliance in hard-to-reach settings requires a comprehensive understanding of the patterns and predictors of protective social behavior.

(https://uasdata.usc.edu/index.php), pending submission and approval of a DUA. All external, county-level data sources are publicly available for download via the New York Times (https://www.nytimes.com/interactive/2021/us/covid-cases.html), Bureau of Labor Statistics (https://www.bls.gov/lau/#cntyaa), and MIT (https://electionlab.mit.edu/data). A linked file in which external data sources are linked to UAS survey data cannot be shared publicly to preserve the anonymity of the respondents. However, this file can be made available to researchers who meet the criteria for access to confidential data (contact uas-l@usc.edu for more information).

**Funding:** Data collection for this study was supported by the University of Southern California, the Bill & Melinda Gates Foundation, and federal funds from the National Institute on Aging (grant 5U01AG054580-03) and the National Science Foundation (grant 2028683). The sponsors had no role in study design, data collection and analysis, decision to publish, or preparation of the manuscript.

**Competing interests:** The authors have declared that no competing interests exist.

Current models of protective behavior fall into two general groups: social-cognitive and social-ecological models. Social-cognitive models emphasize the contribution of individual or intrapersonal factors. One of the most widely-used of these models–the Health Beliefs Model (HBM)–focuses on the role of attitudes and beliefs [3, 4]. Studies of COVID-19 and prior pandemics identify risk perceptions, self-efficacy, and belief in the effectiveness of NPIs as individual-level factors that are positively associated with adherence to protective behavior [5–16]. Some of these studies emphasize the predictive power of perceived self-efficacy and adherence to preventive health behaviors [6, 13]; others emphasize the role of perceived disease risk and effectiveness beliefs in shaping behavioral adherence [7, 12–16].

Beyond health-related attitudes and beliefs, studies have also identified political ideology [12, 17–19] and trusted information sources [14, 20–22] as intrapersonal factors that may be associated with preventive behavior, although the evidence is mixed. Szilagyi et al. [18] and Bruine et al. [19], for example, find that Democrats are systematically more likely than Republicans to adhere to COVID-19 preventive behavior, controlling for demographic and attitudinal factors. Kemmelmeier and Jami [12] observe this as well but find that it is explained primarily by differences in health beliefs. Regarding the predictive power of trusted sources, findings are similarly mixed. Alijanzadeh et al. [14] observe a positive relationship between institutional trust and preventive behavior while Jørgensen et al. [6] posit that interpersonal and institutional trust have limited predictive power compared to health beliefs.

Others note that attitudes and beliefs tend to predict behavioral *intention* better than they do behavior, and intention tends to explain only a portion of the variance observed in health-related behavior [13, 23, 24]. Social-ecological models suggest that environmental factors may fill this explanatory gap. While environmental factors are not entirely absent from social-cognitive models, they are assigned limited explanatory power and rarely receive the same theoretical or empirical attention as health beliefs. The Social-Ecological Model (SEM) differs from these models in that it attributes preventive behavior to the *interplay* between individuals and the wide range of interpersonal (i.e. family, friend), organizational (i.e. work, healthcare, school), and community (i.e. social, political) contexts in which they are embedded [25–28].

According to social-ecological studies of protective behavior and related outcomes [29–31], individuals' interpersonal and community contexts are strongly related to their risk perceptions and adherence to social-distancing and masking behaviors [12, 32–34]. Tunçgenç et al. [32], for example, find that perceived levels of social circle adherence are the strongest predictor of personal adherence to protective measures, controlling for health-related beliefs. High levels of adherence have also been observed in communities with high levels of social capital [35, 36]. On the other hand, Jang's [37] recent social-ecological study of COVID-19 protective behavior in South Korea finds that community- and policy-level factors have relatively limited predictive power compared to individual and interpersonal factors.

In Fig 1, these diverse cognitive and environmental factors are organized into a conceptual framework that aims to synthesize cognitive and ecological models of protective behavior.

Fig 1 points to a range of interrelated factors and potential targets for behavioral intervention, from personal beliefs to the social, economic, and political conditions in which people live and work. When time and resources are limited, an effective pandemic response hinges on the ability of policymakers to anticipate the relative importance of these factors as targets for intervention. To this end, our study makes two important contributions to the research literature.

First, we examine time-*invariant* patterns of protective behavior, or patterns of behavior that are relatively *stable* across time points. To elicit these patterns, we draw on 28 waves of survey data, collected by the Understanding America Study from April 2020 to July 2021 (during the first three waves of the COVID-19 pandemic), and we use cluster analysis to derive

**Cognitive Models**

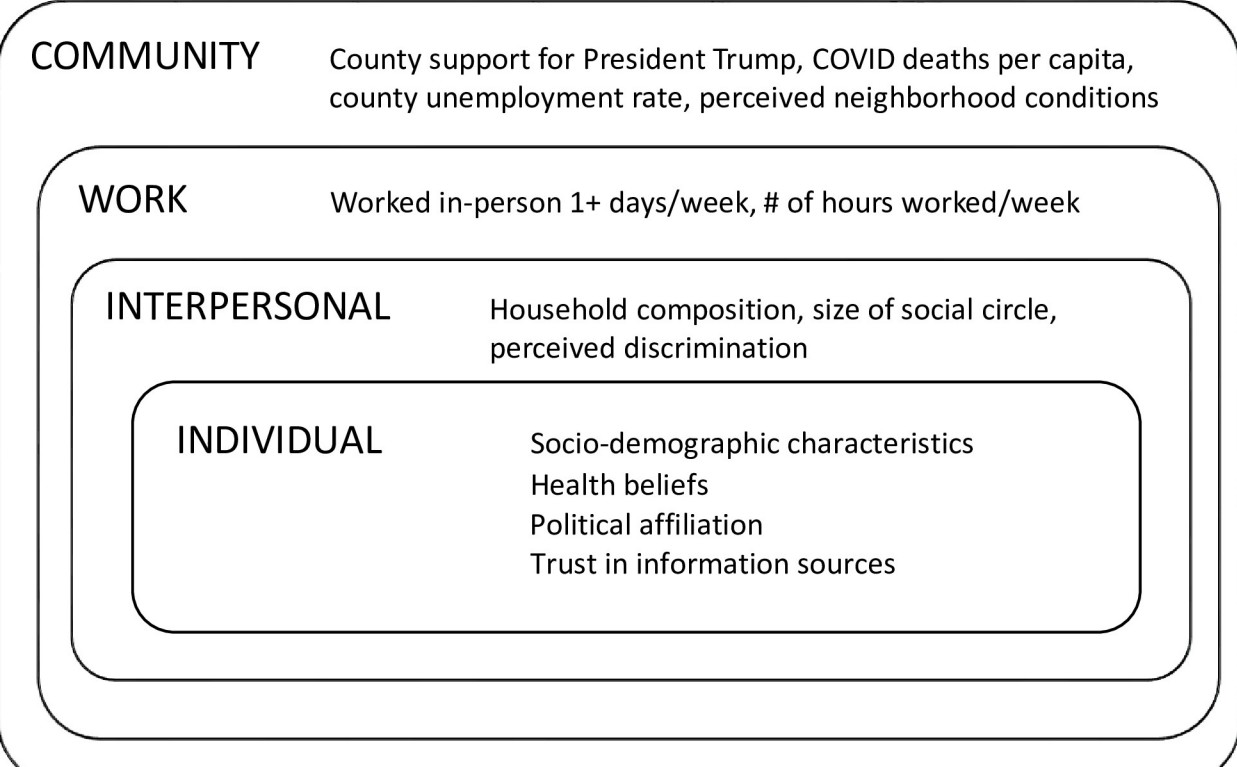

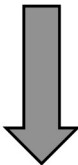

Adherence to Protective Behavior

**Fig 1. Modified conceptual framework and measures, based on social-ecological and social-cognitive models.**

average patterns of adherence to private social-distancing and masking behavior. The benefit of this measurement approach is that it elicits general and enduring patterns of behavior in the population, which are especially useful for anticipating population response to future pandemics. It is a fairly novel approach as well, as most prior studies measure population behavior at a single point in time (e.g. Kleitman et al.'s [38] cluster analysis of behavioral adherence) or they focus on the patterns and predictors of *time-varying* behaviors, or behavioral change (e.g. Schumpe et al.'s [34] longitudinal study of public health behavior).

Second, we use meta-regression analysis to assess the *relative contribution* of individual and environmental factors to the prediction of behavioral adherence. This fills a gap in current research as most studies are designed to identify predictive factors but not necessarily to weigh their predictive value. In contrast, our meta-regression analysis of behavioral adherence during the first three waves of the COVID-19 pandemic reveals substantial variation in the predictive value of cognitive and environmental factors. Health beliefs emerge as by far the single

strongest predictor of behavioral adherence, accounting for 80% of the variance observed. Other factors, while significantly correlated with adherence, have relatively poor predictive power or primarily indirect effects. We conclude that in a public health emergency where time and resources are limited, health beliefs constitute a strategic site for research and intervention.

## Methods

### Study sample

The Understanding America Study (UAS) is a probability-based internet panel of approximately 9,500 non-institutionalized adults residing in the United States. UAS panelists are recruited via address-based sampling, which allows for valid statistical inferences and avoids the coverage problems of convenience web-based panels. UAS panel recruitment and refreshment occurs on an annual basis in successive batches, using two-stage random sampling. As a first step we randomly draw a sample of zip codes, and as a second step we draw addresses within selected zip codes. In this way, every household in the sample has a known probability of selection, which allows for the construction of sample weights. Recruiting households into the UAS is done through a multi-mode approach involving pencil-and-paper, telephone, and Internet modes of response to ensure representativeness of the sample. Households that do not have access to Internet are provided with a tablet and a high-speed Internet connection. From thereon, surveys are conducted over the Internet. All UAS surveys are conducted in English or Spanish and survey respondents receive $20 per 30 minutes of survey time.

From April 1, 2020, to July 6, 2021, 10,279 UAS panelists were invited to participate in a COVID-19 tracking survey called the "Understanding Coronavirus in America" survey [27, 39]; 88% of active UAS panelists consented to participate. Consenting panelists were surveyed biweekly (from April 1, 2020, to February 16, 2021) or every four weeks (from February 17, 2021, to July 20, 2021) about a number of topics related to COVID-19, including their mental and physical health, preventive health behaviors. Each day, on a rolling basis, one-fourteenth (one-twenty-eighth after February 17, 2021) of the panel were invited to complete a given survey wave over the course of two weeks.

The core survey questionnaire was developed by a team of experts in survey methodology, economics, sociology, psychology, and health-related research at the University of Southern California, with input from the Los Angeles County Department of Public Health. A total of 28 surveys were fielded, with a mean completion rate of 75%. Data collection was approved by the University of Southern California Institutional Review Board. All survey data are publicly available online. Analytic samples for this study comprise respondents with complete data (i.e. non-missing values in at least one survey wave) for our predictors and outcomes of interest.

The starting analytic sample comprises 8,616 UAS panel members who participated in at least one wave of the Understanding Coronavirus in America survey and who answered questions about their private masking and social-distancing behavior. The demographic breakdown of the sample can be found in the Total column of Table 1. Select demographic characteristics, presented alongside population benchmarks from the 2020 ACS 5-Year Estimates, can also be found in the S1 Appendix.

Among respondents, 52% are female, 54% are married, 40% are Democrats, 27% are aged 18–34 years, 18% are aged 65+ years, 61% are non-Hispanic White, 18% are Hispanic, 12% are non-Hispanic Black, and 6% are non-Hispanic Asian. The socio-economic composition of the sample is 37% high school degree or less, 35% Bachelor's degree or more, 28% household income of less than $30,000, and 23% household income of $100,000 or more. These percentages largely mirror population benchmarks. Survey weights are also included in all of our

**Table 1. Demographic summary by adherence cluster.**

| | Total | High Compliance | Moderate Compliance | Low Compliance | P |
|---|---|---|---|---|---|
| | (N = 8,616) | (N = 3,766) | (N = 3,287) | (N = 1,563) | |
| Age | | | | | <0.001 |
| 18–34 | 2,363 (27.4%) | 923 (24.5%) | 938 (28.5%) | 502 (32.2%) | |
| 35–49 | 2,572 (29.9%) | 1,004 (26.7%) | 1,047 (31.9%) | 521 (33.4%) | |
| 50–64 | 2,125 (24.7%) | 1,025 (27.3%) | 758 (23.1%) | 342 (21.9%) | |
| 65+ | 1,552 (18.0%) | 811 (21.6%) | 544 (16.6%) | 197 (12.6%) | |
| Gender | | | | | <0.001 |
| Female | 4,502 (52.23%) | 2,042 (54.2%) | 1,712 (52.1%) | 748 (47.8%) | |
| Male | 4,113 (47.7%) | 1,724 (45.8%) | 1,574 (47.9%) | 815 (52.2%) | |
| Race/Ethnicity | | | | | <0.001 |
| White | 5,254 (61.0%) | 1,908 (50.7%) | 2,125 (64.7%) | 1,221 (78.1%) | |
| Hispanic | 1,521 (17.7%) | 806 (21.4%) | 511 (15.6%) | 204 (13.0%) | |
| Black | 1,019 (11.8%) | 650 (17.3%) | 312 (9.5%) | 57 (3.7%) | |
| Asian | 478 (5.6%) | 262 (7.0%) | 199 (6.1%) | 17 (1.1%) | |
| Other | 337 (3.9%) | 137 (3.6%) | 136 (4.2%) | 64 (4.1%) | |
| Education | | | | | <0.001 |
| High School or Less | 3,198 (37.1%) | 1,424 (37.8%) | 1,161 (35.4%) | 613 (39.2%) | |
| Some College | 2,445 (28.4%) | 1,071 (28.4%) | 906 (27.6%) | 468 (29.9%) | |
| B.A. or More | 2,970 (34.5%) | 1,271 (33.8%) | 1,216 (37.0%) | 483 (30.9%) | |
| Household Income | | | | | <0.001 |
| Less than $30,000 | 2,412 (28.0%) | 1,291 (34.3%) | 811 (24.7%) | 310 (19.9%) | |
| $30,000-$59,999 | 2,291 (26.6%) | 972 (25.9%) | 886 (27.0%) | 433 (27.7%) | |
| $60,000-$99,999 | 1,961 (22.8%) | 732 (19.5%) | 796 (24.3%) | 433 (27.7%) | |
| $100,000 or more | 1,940 (22.5%) | 767 (20.4%) | 787 (24.0%) | 386 (24.7%) | |
| Marital Status | | | | | 0.005 |
| Married | 4,611 (53.5%) | 1,946 (51.7%) | 1,767 (53.8%) | 898 (57.4%) | |
| Never Married | 2,350 (27.3%) | 1,058 (28.1%) | 898 (27.4%) | 394 (25.2%) | |
| Other | 1,651 (19.2%) | 760 (20.2%) | 619 (18.8%) | 272 (17.4%) | |
| Region | | | | | <0.001 |
| Northeast | 1,494 (17,4%) | 648 (17.3%) | 590 (18.2%) | 256 (16.4%) | |
| South | 3,314 (38.8%) | 1,563 (41.8%) | 1,178 (36.3%) | 573 (36.7%) | |
| Midwest | 1,723 (20.2%) | 594 (15.9%) | 702 (21.6%) | 427 (27.4%) | |
| West | 2,017 (23.6%) | 931 (24.9%) | 781 (24.0%) | 305 (19.5%) | |
| Political Affiliation | | | | | <0.001 |
| Democrat | 2,464 (40.2%) | 1,388 (51.0%) | 930 (39.4%) | 146 (13.9%) | |
| Republican | 2,224 (36.3%) | 632 (23.2%) | 921 (39.1%) | 671 (63.9%) | |
| Other | 1,445 (23.6%) | 704 (25.8%) | 508 (21.5%) | 233 (22.2%) | |

analyses to account for small differences between the demographic composition of our sample and that of the U.S. population.

## Outcome measures

Our outcome of interest is average adherence to private social-distancing and private masking behavior. This outcome is measured using responses to a series of social-distancing and masking behavior questions from the COVID tracking survey. These questions were developed internally, by the UAS survey team, based on CDC recommendations for COVID-19 prevention [40]. Published analyses of the data suggest these measures are reliable and valid, as they

have relatively stable relationships over time [41], are responsive to changes in public health guidelines [15], and they relate to other variables, like health-related beliefs [15] and political affiliation [19], in expected ways. These measures are detailed below.

In waves 1–28 of the COVID survey, respondents were asked if they had participated in 16 different social activities in the last seven days. Response options included "(1) Yes," "(2) No," and "(3) Unsure." Of the activities listed, we use responses to the following four to measure adherence to private social-distancing behavior (mean and standard deviation are included in parentheses): (1) "Gone to a friend, neighbor, or relative's residence (that is not your own)" (mean: 1.59; sd: 0.34), (2) "Had visitors such as friends, neighbors or relatives are your residence" (mean: 1.59; sd: 0.34) (3) "Attended a gathering with more than 10 people, such as a reunion, wedding, funeral, birthday party, or religious service" (mean: 1.86; sd: 0.24) and (4) "Had close contact (within 6 feet) with people who do not live with you" (mean: 1.40; sd: 0.35). We selected these four activities because they have been identified by government agencies as private social situations with a high risk of COVID transmission [40, 42]. We combined the first two activities into a single item: "household visits." We refer to the third item as "social gatherings" and the fourth as "close contact."

A follow-up to the above was added to waves 7–28 to assess adherence to private masking behavior. For each activity that received a participation response of "Yes," respondents were asked ". . .how often, if ever, you wore a mask or face covering," with the following as response options: (1) Unsure, (2) Never, (3) Rarely, (4) Sometimes, (5) Most of the time, (6) Always. For social-distancing item 1, the average masking frequency score is 3.54 (sd: 1.30); for item 2, the average masking frequency score is 3.26 (sd: 1.24); for item 3, the average masking frequency score is 4.25 (sd: 1.38); for item 4, the average masking frequency score is 4.44 (sd: 1.12).

We applied the following coding scheme to measure adherence to private social-distancing and masking behavior. In waves 1–6, our coding scheme was dichotomous and based on social-distancing behavior. For each activity of interest (household visits, social gatherings, close contact), respondents received a code of "1" (denoting adherence) if their participation response was "No" or "Unsure" and a code of "0" (denoting non-adherence) if their participation response was "Yes."

In waves 7–28, our coding scheme was ordinal and based on both social-distancing and masking behavior. Respondents received a code of "2" (denoting high adherence) if their participation response was "No" or "Unsure"; they received a code of "1" (denoting moderate adherence) if their participation response was "Yes" AND their mask frequency response was "Always" or "Most of the time"; they received a code of "0" (denoting non-adherence) if their participation was "Yes" AND their mask frequency response was "Sometimes," "Rarely," "Never," or "Unsure."

After coding adherence, we performed a cluster analysis to group respondents on the basis of their adherence values across all 28 survey waves. Hierarchical complete linkage clustering was used, with a distance matrix derived from Euclidean distances along the first principal component of the time-averaged adherence measures. A 3-cluster solution was extracted, with one cluster consistently having the highest adherence measure means ("high adherence"), one consistently having the lowest ("low adherence"), and one consistently in between ("moderate adherence").

### Individual-level predictors

As presented in Fig 1, our individual-level predictor domains include socio-demographic characteristics, political affiliation, trusted information sources, and health beliefs. To measure socio-demographic characteristics, we draw on indicators of race and ethnicity (non-Hispanic

white, non-Hispanic Black, non-Hispanic Asian, Hispanic, and non-Hispanic other), gender (male, female), age (18–34 years, 35–49 years, 50–64 years, 65+ years), education (high school degree or less, some college, Bachelor's degree or more), household income (less than $30,000, $30,000–59,999, $60,000–99,999, $100,000+), marital status (married, never married, other), at least one chronic health condition (diabetes, cancer (other than skin cancer), heart disease, high blood pressure, asthma, chronic lung disease, kidney disease, autoimmune disorder, mental health condition, obesity), and health insurance status (yes, no). All measures except the last two (chronic conditions and health insurance status) were collected and updated by UAS respondents on a quarterly basis through the UAS *My Household* survey. Measures of health insurance status and chronic health conditions were collected in waves 1–28 of the COVID tracking survey.

To measure political affiliation, we use the political party affiliation (Democrat, Republican, and Other) reported by respondents in UAS 318, a post-election poll distributed to full UAS panel from November 4, 2020, to December 15, 2020 (81% survey completion rate).

To measure trusted information sources, we rely on a question series included in waves 1, 7, 19–28 of the COVID survey. Respondents were asked to rate (on a 4-point Likert scale: Do not trust, Trust somewhat, Trust mostly, Trust completely) their level of trust in 22 sources of information about COVID-19. These questions mirror those used in prior studies of trust in health information sources [5, 43–46]. For parsimony, we grouped information sources into the following categories: national mainstream media, national left-leaning media, national right-leaning media, local media, CDC/HHS, physician, local public health officials, social media, and friends/family. For each category, we averaged responses across information sources within the category and then averaged across waves.

To measure health beliefs, we draw on the following indicators, collected in waves 1–28 of the COVID survey unless otherwise noted: (1) perceived susceptibility, measured as perceived risk of getting COVID (visual linear scale: 0%. . .100%), perceived risk of vaccinated individuals getting COVID (visual linear scale: 0%. . .100%; waves 19–28), and perceived safety of visiting others in home (extremely safe, somewhat safe, somewhat unsafe, extremely unsafe, unsure), (2) perceived severity, measured as perceived risk of death if infected with COVID (visual linear scale: 0%. . .100%), (3) perceived benefits, measured as self-reported (dis)agreement (5-point Likert scale: strongly disagree. . .strongly agree) that face masks "keep me safe," "keep others safe," "are not needed because I am not infected," "are not needed when I am with other people who are healthy," are not needed because "I keep enough distance" (waves 7, 9–28), (4) perceived barriers, measured as self-reported (dis)agreement that face masks are dangerous, threatening to others, unaffordable, uncomfortable (waves 7, 9–28), and (5) perceived self-efficacy, measured as self-reported frequency of feeling unable to control important things in life (5-point Likert scale: never. . .very often; from the validated Perceived Stress Scale 4 [47]).

All risk perception questions used validated visual linear response scale ranging from 0% to 100% [48]. Published analyses confirm that the risk perception measures relate to protective behavior measures in expected ways, especially after COVID-19 was declared a public health emergency [15, 19]. Studies also indicate that anti-masking attitudes correlate strongly with negative attitudes towards COVID-19 vaccination, political conservatism, and resistance to social-distancing measures [49]. See S2 Appendix for a full description of the survey measures used in this study.

## Environmental predictors

Our environmental predictor domains of interest are interpersonal context, work context, and community context. To account for interpersonal context, we draw on indicators of household

composition (household members age 65+, children age 0–4, children age 5–18 –from the UAS *My Household* survey), number of close friends/family (<10, 10–20, 21–37, 38+—from waves 1–5, 7, 9–28 of the COVID survey), self-reported experiences of everyday discrimination (validated everyday discrimination scale, short version [50], collected in waves 6–7, 9–28), and self-reported experiences of COVID-associated discrimination (adapted from the validated everyday discrimination scale, collected in waves 1–5, 7, 9–28). To account for <u>work context</u>, we rely on indicators of respondents' work conditions, including the number of hours worked in the past week and whether or not they worked in-person for at least one day in the last week, collected in waves 1–28. Non-workers received a value of "0" for these measures.

Lastly, to account for <u>community context</u>, we draw on validated measures of perceived neighborhood disorder [51] ((dis)agreement with statements about neighborhood cleanliness, crime, vandalism, loitering–collected in waves 1–28 of the COVID survey), county-level administrative indicators of the COVID death rate (county-level deciles of COVID deaths per capita, published by the *New York Times*), unemployment rate (county-level 2020 unemployment rate, published by the *Bureau of Labor Statistics*), and support for presidential candidate Donald Trump (county-level deciles of % votes for Donald Trump in 2020 presidential election, published by *MIT)*.

## Statistical analysis

Our analytic strategy proceeded in three phases. First, we performed a cluster analysis to group respondents on the basis of their time-invariant patterns of self-reported adherence to private social-distancing and masking behavior. Second, we used ordinal logistic regression models to estimate the predictive value of individual predictors with respect to cluster membership, fitting separate models for each of the following predictor domains.

The predictive performance of each model is summarized in terms of concordance *C* statistics, which can be interpreted as the proportion of respondent pairs correctly ranked by the model in terms of adherence. Finally, we fitted a meta-regression model of cluster membership, representing each domain by the predicted probabilities of high adherence derived from the corresponding model. Ordinal logistic regression was used for the meta-regression analysis, with domain-specific predicted probabilities modeled on the log odds scale.

All analyses accounted for survey sampling weights and used two-sided 0.05 significance levels (SAS v. 9.4).

## Results

### Sample description

Our starting sample comprises 8,616 UAS panel members who participated in at least one wave of the Understanding Coronavirus in America survey and who answered questions about their private masking and social-distancing behavior. Regression model sample sizes vary according to the number of respondents who have complete data (i.e. non-missing values in at least one survey wave) for the outcomes and predictors of interest in a given model. Domain-specific regression model sample sizes (reported in S3 Appendix) range from 6,040 (socio-demographic characteristics model) to 8,229 (work context). The meta-model sample size is 5,364, which reflects the number of respondents with complete data on every measure.

### Patterns of protective social behavior

The results of our cluster analysis reveal three distinct patterns of adherence to protective behavior in private settings: high adherence (44% of respondents), moderate adherence (38%),

and low adherence (18%) behavior. Mean scores for each activity and cluster can be found in the S1 Fig.

As shown in Table 1, demographic characteristics are strongly related to cluster membership. Respondents who are under age 65, male, non-Hispanic white, high-income, lower education, or Republican are more likely to be classified as low adherence than respondents who are aged 65+, female, Hispanic or non-white, low-income, higher educated, or a Democrat, respectively.

## Predictors of protective social behavior

The results of our ordinal logistic regression models are presented in Fig 2. Prediction models are ranked in order of their C-statistic, which summarizes the predictive performance of each model. Health beliefs are the strongest predictor of adherence to protective social behavior. With a C-statistic of 0.80, a model using health beliefs alone correctly ranks pairs of respondents in terms of adherence 80% of the time. All other predictors do a notably worse job of prediction, with C-statistics ranging from 0.58 (work context) to 0.64 (Informational trust, political affiliation, socio-demographic characteristics). Individual-level domains emerge as only marginally more predictive than environmental domains.

The statistically significant correlates of adherence within each domain are presented in Fig 2 (see S3 Appendix for detailed regression models). There are a number of correlates that move in expected directions. For example, perceived risk of death and the perception of masks as beneficial to health are positively associated with adherence level. Trust in national and left-leaning media and rates of county-level unemployment are both positively associated with

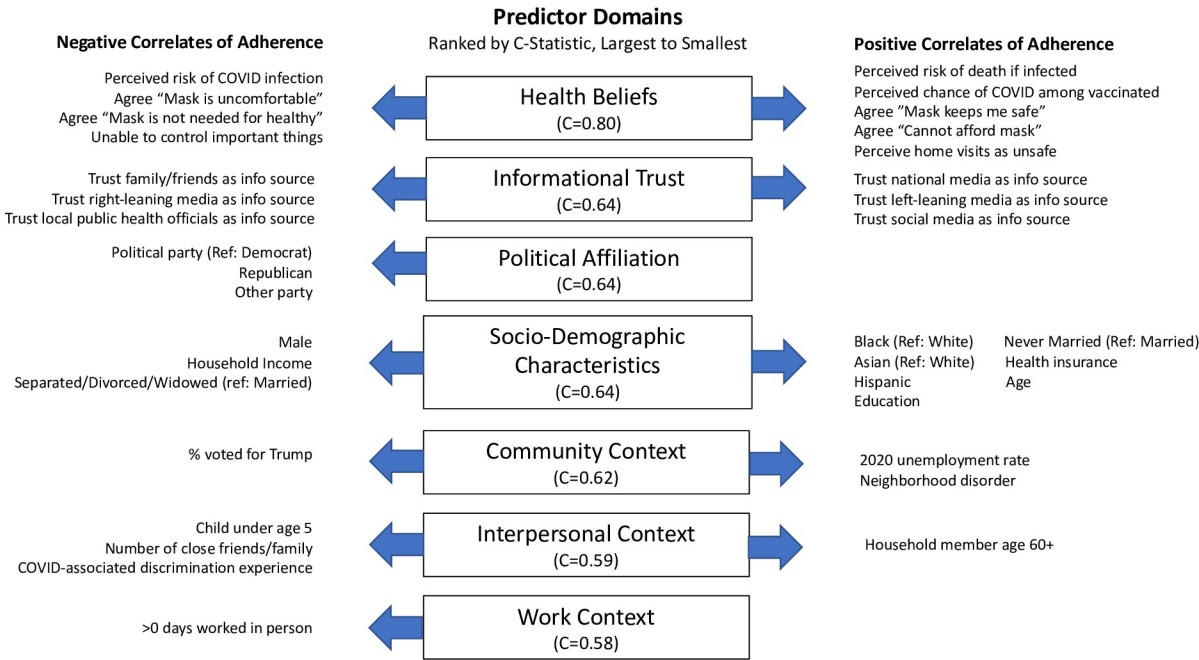

**Fig 2. Predictors of adherence to protective social behavior.** *Note*: C-statistics summarize the predictive performance of each prediction model. A C-statistic can be interpreted as the proportion of respondent pairs correctly ranked by the model in terms of adherence. For example, a C-statistic of 0.80 indicates the predictor domain correctly ranks pairs of respondents in terms of adherence 80% of the time. A C-statistic of 0.50 indicates the predictor domain is no more predictive than random chance. See S3 Appendix for all models used to identify statistically significant correlates and for the variables included in each model but omitted from this figure because they were not statistically significant.

adherence, while affiliation with the Republican party, in-person work, and rates of county-level support for President Trump in the 2020 election are negatively associated with adherence.

There are a few significant correlates that move in unexpected directions. For example, there is a significant negative association between perceived risk of COVID infection and adherence and a significant positive association between inability to afford a mask and adherence. Within each domain, there are also measures that do *not* exhibit a statistically significant relationship to adherence. In the health beliefs domain, three of the four perceived barrier measures–i.e., (dis)agreement that masks are dangerous, uncomfortable, or threatening–*do not* have a statistically significant relationship to adherence. Three of the five perceived benefit measures–i.e. (dis)agreement that masks keep others safe, not necessary because not infected, or not necessary because keep enough distance–are also non-significant. In the informational trust domain, trust in local media, federal health agencies (CDC and HHS), and physicians are non-significant. In the work context domain, in-person work has a significant relationship to adherence but number of hours worked does not. Lastly, in the community context domain, COVID deaths per capita do not have a significant relationship to adherence. Importantly, the significant and non-significant associations detailed above may be due to demographic or other confounders not included in each domain-specific regression model. We account for confounders in our meta-regression analysis.

Meta-regression results can be found in Fig 3, which plots the percentage of each domain's predictive value that is retained after controlling for all other domains. On the x-axis, each domain is plotted in order of its C-statistic to denote the domain's predictive value prior to controls. See S3 Appendix for odds ratios and standard errors from the regression model.

Results show that health beliefs retain 94% ($p < 0.001$) of their predictive value, which means only 6% is explained away by other domains. This percentage-retained is much higher than the percentage retained by other individual and contextual domains. After health beliefs, interpersonal context retains the highest percentage of its predictive value, at 87% ($p < 0.001$), although its predictive performance is quite poor, as indicated by its C-statistic of 0.59. Community context, work context, and socio-demographic characteristics retain 52% ($p < 0.001$), 34% ($p < 0.001$), and 23% ($p < 0.001$) of their predictive value after adjustment, respectively. Most notably, political affiliation retains 6% ($p = 0.166$) of its predictive value and informational trust retains 0% ($p = 0.992$), meaning essentially all of the predictive value of political affiliation and informational trust is explained by their relationship to other domains, mainly health beliefs.

The C-statistic for the entire meta-model (0.82) is not much higher than the C-statistic generated by health beliefs alone (0.80). This suggests individual and contextual domains contribute very little to the prediction of adherence above and beyond their relationship to health beliefs.

## Discussion

Two key findings emerge from this analysis of protective behavior in private social settings. First, patterns of adherence fall into three categories marked by high, moderate, or low levels of social-distancing and masking behavior, with under half (44%) of respondents exhibiting consistently high levels of adherence to protective measures. This differs from the pattern of adherence observed by Kleitman et al. [38] in their latent class analysis of protective behavior during the first wave of the COVID-19 pandemic. Based on surveys distributed in April and May 2020 to a non-random sample of participants from Australia, Canada, the U.S., and the U.K., Kleitman and colleagues observe just two behavioral patterns–compliance and non-

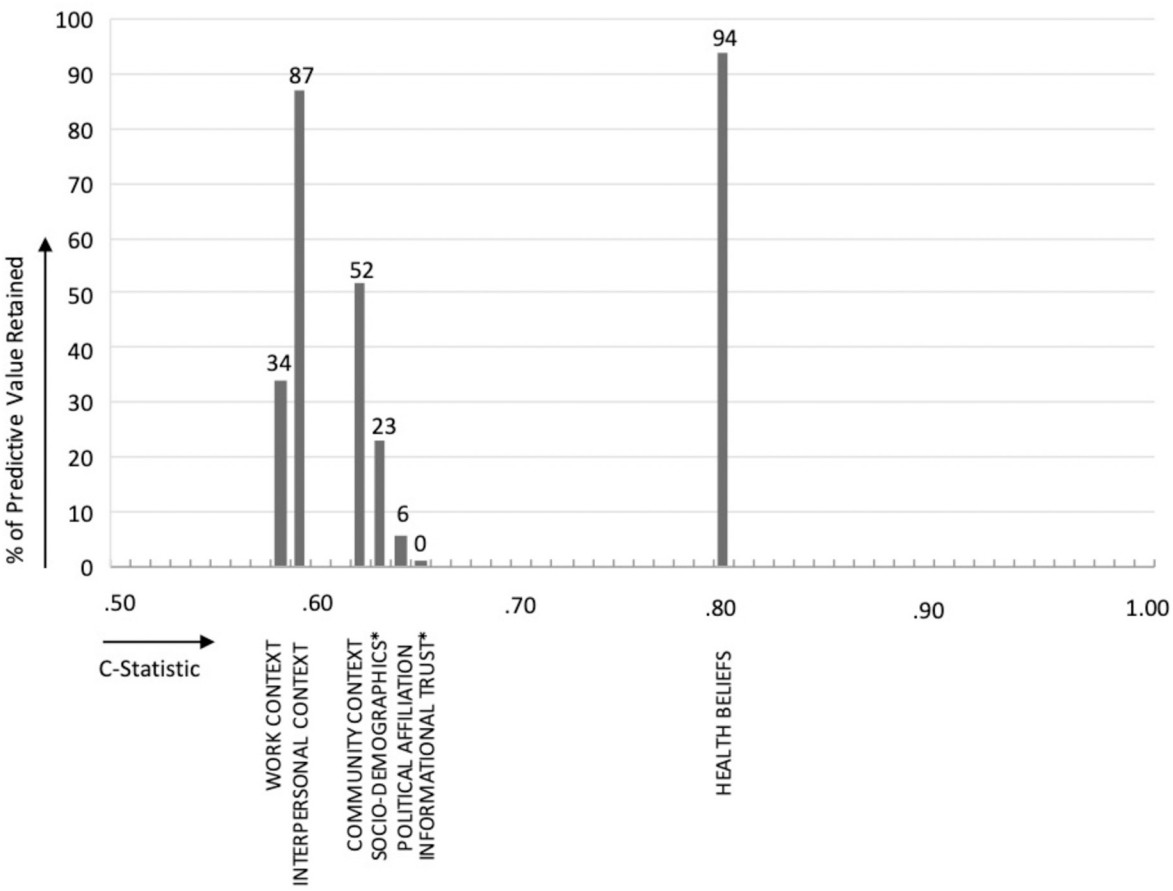

**Fig 3. Predictive value retained in meta-regression model, by predictor domain and C-statistic.** *Note*: This figure reports the percentage of each domain's predictive value (measured as the predicted probability of high adherence associated with the modification of that domain) that is retained after controlling for all other domains. For example, health beliefs retain 94% of their predictive value after adjusting for other domains, which means that 6% of the predictive value of health beliefs is explained away by other domains. On the x-axis, each domain is plotted in order of its C-statistic. C-statistics represent the domain's original predictive performance prior to controls. A C-statistic of 0.50 is the same as random chance. *These predictors have the same C-statistic as political affiliation (0.64).

compliance–with 90% of respondents classified as compliant. However, this high rate of compliance was unique to the early days of the pandemic and its generalizability to the U.S. population is unclear given the sampling strategy used. Our analysis, in contrast, draws on a representative sample of U.S. residents surveyed over the course of three pandemic waves.

Second, in support of prior research on the HBM [52], health beliefs emerge from our analysis as the single strongest predictor of adherence to protective social behavior. Alone, they predict the largest share of observed adherence (C = 0.80) and they have the strongest direct relationship to adherence, retaining 94% of their predictive value after adjustment for other domains. Compared to health beliefs, all other contextual and individual-level predictors, which include socio-demographic factors, political affiliation, informational trust, interpersonal context, work context, and community context, have either poor predictive power or primarily indirect associations.

Political affiliation and informational trust, for example, are moderately predictive of behavioral adherence, with a C-statistic of 0.64, but their relationship to behavior is entirely indirect–primarily explained by their relationship to health beliefs. Interpersonal context, in contrast, exhibits a direct relationship to adherence–retaining 87% of its predictive value after

adjustment for other domains–but it has very poor predictive power (C = 0.59). Socio-demographic factors, work context, and community context fall somewhere in the middle, with weak to moderate predictive power (ranging from 23%-52% predictive value retained after controls) and weak to moderate direct effects (ranging from C = 0.58 to C = 0.64).

In line with prior studies [6, 12, 53, 54] our results show plainly that political affiliation and informational trust *do* influence protective behavior but only insofar as they influence health beliefs. Similarly, Kemmelmeier and Jami [12] find in their study of U.S. masking behavior that the relationship between political affiliation and masking behavior is explained almost entirely by beliefs about behavioral effectiveness. Likewise, in their study of protective behavior in Western Europe, Jørgensen et al. [6] find that interpersonal and institutional trust have limited predictive power above and beyond health-related beliefs.

Our results also share some similarities and a number of differences with recent social-ecological studies of adherence to COVID-19 protective measures. In a social-ecological analysis of protective-behavioral adherence in South Korea, Jang [37] uses cross-sectional survey data, collected in December 2020, to assess the relative importance of multi-level factors in predicting adherence. Based on a comparison of R-squared measures across models with differing combinations of predictors, Jang concludes that health-related beliefs have relatively high predictive value, followed by interpersonal context; community and policy factors, in contrast, have relatively low predictive value. Substantively, our results and conclusions are quite similar to Jang's, albeit in a different national context. From a methodological standpoint, however, we improve upon Jang's study by (1) analyzing longitudinal survey data collected over the first three waves of the pandemic and (2) using a meta-regression analysis approach to evaluate and compare the predictive value of cognitive and environmental factors.

Other social-ecological studies of adherence emphasize the importance of interpersonal and community context, although few actually analyze the *relative* contribution of these factors to adherence [28, 31–33, 35, 36]. Tunçgenç et al. are one of the few to compare contextual to individual-level predictors of adherence. Notably, they observe that attitudinal measures (i.e. approval of protective measures of perceived infection risk), while strongly predictive of personal adherence, are exceeded in importance by social circle adherence. These results suggest interpersonal context may carry more weight than what we capture in our own study. It is certainly a limitation of our study that we do not account for social circle adherence as a measure of interpersonal context, only social circle size. That said, the internal and external validity of Tunçgenç's results are unclear as they are based on self-report measures (i.e. *perceived* social circle adherence) collected in the early days of the pandemic (April 2020), from a non-probability sample of European university students.

A few limitations to this analysis should be acknowledged. First, our results describe protective behavior in *private interpersonal settings*. We focus on these settings because they have been hotspots for the community spread of COVID-19 [42, 55] but the private nature of these settings may explain why the effect of community context in our model is primarily mediated by individual-level factors. In public settings, we might expect a different pattern than this to unfold, where communities directly influence behavior without necessarily influencing beliefs.

Second, as we have already alluded to, this analysis includes a limited set of measures of organizational and interpersonal context. While we were able to account for the size of respondents' social networks in this study, we could not account for network composition or behavior, although we expect this to be correlated with community and socio-demographic characteristics. Furthermore, while we were able to account for work conditions, we could not account for other organizational contexts in which respondents are embedded. Third, our results reflect average patterns and predictors of adherence to protective behavior during the COVID-19 pandemic, which may or may not mirror patterns and predictors at a single point

in time or in other public health emergencies. For example, research suggests that adherence to protective behavior guidelines tends to decline over time–a phenomenon called "behavioral fatigue" [56].

## Conclusion

The results of this study have implications both for public health strategies to mitigate the spread of COVID-19 and for the general challenge of disease mitigation in public health emergencies. Results show clearly that health beliefs drive adherence to protective measures in private social settings; other individual-level and contextual factors carry only modest predictive value or their relationship to behavior is largely mediated by their relationship to health beliefs.

As detailed in the discussion of results, our study limitations include a focus on private interpersonal settings, time-invariant patterns of protective behavior, and a restricted set of measures of organizational and interpersonal context. To address the first of these limitations, one promising direction for future research will be to expand this analysis to include other types of protective behavior such as COVID-19 vaccination and public-facing NPIs (e.g. social-distancing and masking in grocery stores, airplanes, etc.). This expanded analysis would lend important insight into how different types of preventive behavior relate to one another and to cognitive and environmental factors.

Current research and policy efforts would also benefit from longitudinal studies of protective-behavioral trajectories [57], including the prevalence, patterns, and predictors of behavioral fatigue [56]. While this study focused on time-invariant behavioral patterns, time-varying patterns demand our attention as well, and it is well-documented that adherence can diminish over time in certain segments of the population. Furthermore, the generalizability of this study to other public health emergencies is unknown. As the COVID-19 research literature continues to grow, meta-analyses and comparative studies will play a critical role in distinguishing the behavioral patterns and predictors common to all public health emergencies from those unique to the COVID-19 pandemic.

A final word of caution: It is important not to conclude from the results of this study that indirectly predictive factors, like informational trust or community context, are unimportant or undeserving of further study. To the contrary, our results suggest these factors *do* matter, primarily *because* of their relationship to health beliefs. This means that local communities and trusted messengers may have minimal impact on private protective behavior if their messages are misaligned with, or fail to change, people's beliefs. A successful mitigation strategy for private social settings will depend upon the ability of trusted sources, messengers, policy-makers, and communities to influence health beliefs. A critical direction for future research will therefore be the effective design of interventions at the level of communities, institutions, and trusted messengers that aim to change health beliefs.

## Supporting information

**S1 Appendix. Sample description.**
(DOCX)

**S2 Appendix. Select measure descriptions.**
(DOCX)

**S3 Appendix. Regression models.**
(DOCX)

**S1 Fig. Adherence item means, by adherence cluster.**
(TIF)

# Acknowledgments

We thank Rashmi Shetgiri, MD, MSHS, MSCS, and Paul Simon, MD, MPH, of the Los Angeles County Department of Public Health as well as Jeanne R. Delgado, MD, MPH, of the National Clinician Scholars Program, Division of General Medicine & Health Services Research, at the University of California, Los Angeles, for their input on study design and interpretation of results.

# Author Contributions

**Conceptualization:** Kyla Thomas, Peter G. Szilagyi, Sitaram Vangala, Rebecca N. Dudovitz, Megha D. Shah, Nathalie Vizueta, Arie Kapteyn.

**Formal analysis:** Sitaram Vangala.

**Funding acquisition:** Arie Kapteyn.

**Investigation:** Kyla Thomas, Arie Kapteyn.

**Methodology:** Sitaram Vangala.

**Writing – original draft:** Kyla Thomas.

**Writing – review & editing:** Kyla Thomas, Peter G. Szilagyi, Sitaram Vangala, Rebecca N. Dudovitz, Megha D. Shah, Nathalie Vizueta, Arie Kapteyn.

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
