## [Decision Letter · Decision Letter 0]

6 Dec 2022

PONE-D-22-23931Behind Closed Doors: Protective Social Behavior During the COVID-19 PandemicPLOS ONE

Dear Dr. Thomas,

Thank you for submitting your manuscript to PLOS ONE. After careful consideration, we feel that it has merit but does not fully meet PLOS ONE’s publication criteria as it currently stands. Therefore, we invite you to submit a revised version of the manuscript that addresses the points raised during the review process.

Please read carefully comments from Reviewers 1 and 2 and approach all of them.

We look forward to receiving your revised manuscript.

Kind regards,

Celia Andreu-Sánchez

Academic Editor

PLOS ONE

Journal Requirements:

2. Please provide additional details regarding participant consent. In the ethics statement in the Methods and online submission information, please ensure that you have specified what type you obtained (for instance, written or verbal, and if verbal, how it was documented and witnessed). If your study included minors, state whether you obtained consent from parents or guardians. If the need for consent was waived by the ethics committee, please include this information

Reviewers' comments:

Reviewer's Responses to Questions

**Comments to the Author**

1. Is the manuscript technically sound, and do the data support the conclusions?

Reviewer #1: Partly

Reviewer #2: Yes

2. Has the statistical analysis been performed appropriately and rigorously? 

Reviewer #1: No

Reviewer #2: Yes

3. Have the authors made all data underlying the findings in their manuscript fully available?

Reviewer #1: Yes

Reviewer #2: No

4. Is the manuscript presented in an intelligible fashion and written in standard English?

Reviewer #1: Yes

Reviewer #2: Yes

5. Review Comments to the Author

Reviewer #1: The submission entitled "Behind Closed Doors: Protective Social Behavior During the COVID-19 Pandemic" is a timely topic. The strengths of the submission included (i) the large sample size; (ii) the importance of the topic; and (iii) a longitudinal study design. However, there are some improvements that should be done. Please see my specific comments below.

1. I feel that the authors did not introduce the protective behaviors during COVID-19 pandemic thoroughly. The current literature has ample information; however, the references used in the Introduction are mostly focus on vaccine or not about COVID-19. The authors may consider the following references discussing the protective behaviors during COVID-19 pandemic to strengthen their Introduction.

Liu, E., & Arledge, S. (2022). Individual characteristics and demographics associated with mask wearing during the COVID-19 pandemic in the United States. Asian Journal of Social Health and Behavior, 5, 3-9.

Prasiska, D. I., MUHLIS, A. N., & Megatsari, H. (2022). Effectiveness of the emergency public activity restrictions on COVID-19 epidemiological parameter in East Java Province, Indonesia: An ecological study. Asian Journal of Social Health and Behavior, 5, 33-39.

Chung, G. K.-K., Strong, C., Chan, Y.-H., Chung, Y.-N., Chen, J.-S., Lin, Y.-H., Huang, R.-Y., Lin, C.-Y., Ko, N.-Y. (2022). Psychological distress and protective behaviors during the COVID-19 pandemic among different populations: Hong Kong general population, Taiwan healthcare workers, and Taiwan outpatients. Frontiers in Medicine, 9, 800962.

Alijanzadeh, M., Ahorsu, D. K., Mahmoudi, N., Majd, N. R., Alimoradi, Z., Griffiths, M. D., Lin, C.-Y., Liu, H.-K., & Pakpour, A. H. (2021). Fear of COVID-19 and trust in the healthcare system mediates the association between individual’s risk perception and preventive COVID-19 behaviours among Iranians. International Journal of Environmental Research and Public Health, 18, 12146.

Pakpour, A. H., Liu, C.-h., Hou, W.-L., Chen, Y.-P., Li, Y.-P., Kuo, Y.-J., Lin, C.-Y., Scarf, D. (2021). Comparing fear of COVID-19 and preventive COVID-19 infection behaviors between Iranian and Taiwanese older people: Early reaction may be a key. Frontiers in Public Health, 9, 740333.

Chang, K.-C., Strong, C., Pakpour, A. H., Griffiths, M. D., & Lin, C.-Y. (2020). Factors related to preventive COVID-19 infection behaviors among people with mental illness. Journal of the Formosan Medical Association, 119(12), 1772-1780.

Lin, C.-Y., Imani, V., Majd, N. R., Ghasemi, Z., Griffiths, M. D., Hamilton, K., Hagger, M. S., & Pakpour, A. H. (2020). Using an Integrated Social Cognition Model to Predict COVID-19 Preventive Behaviours. British Journal of Health Psychology, 25(4), 981-1005.

2. I think that the authors may want to discuss the issue of behavioral fatigue in the preventive behaviors of COVID-19.

Harvey N. (2020). Behavioral Fatigue: Real Phenomenon, Naïve Construct, or Policy Contrivance?. Frontiers in psychology, 11, 589892.

3. I wonder if the authors can provide information regarding whether the measures are reliable and valid.

4. I wonder why the authors used cluster analysis instead of latent class analysis to classify the participants into different groups. I believe that using latent class analysis can tackle the issue of repeated measures and provide a more accurate clasification.

5. Can the authors provide some information regarding the sample representativeness?

6. Again in the Discussion, I feel that the authors did not compare their results with the existing evidence on the protective behaviors during COVID-19 pandemic.

Reviewer #2: Please see the attached file for the detailed comments and suggestions

Overall, the paper is interesting but it needs further improvement and clarification as to how this study is needed (contribution to the existing knowledge is unclear due to poor explanation of research gaps).

6. PLOS authors have the option to publish the peer review history of their article (what does this mean?). If published, this will include your full peer review and any attached files.

Reviewer #1: No

Reviewer #2: No

---

## [Author Response · Author response to Decision Letter 0]

31 Mar 2023

Dear Reviewers: Thank you for your careful reading of the manuscript and for your suggestions for revision. Each reviewer comment is copied/pasted 

below and followed by our response. You can also find our revisions highlighted in yellow in the file “Revised Manuscript with Track Changes.”

REVIEWER #1

1. I feel that the authors did not introduce the protective behaviors during COVID-19 pandemic thoroughly. The current literature has ample information; however, the references used in the Introduction are mostly focus on vaccine or not about COVID-19. The authors may consider the following references discussing the protective behaviors during COVID-19 pandemic to strengthen their Introduction.

Liu, E., & Arledge, S. (2022). Individual characteristics and demographics associated with mask wearing during the COVID-19 pandemic in the United States. Asian Journal of Social Health and Behavior, 5, 3-9.

Prasiska, D. I., MUHLIS, A. N., & Megatsari, H. (2022). Effectiveness of the emergency public activity restrictions on COVID-19 epidemiological parameter in East Java Province, Indonesia; An ecological study. Asian Journal of Social Health and Behavior, 5, 33-39.

Chung, G. K.-K., Strong, C., Chan, Y.-H., Chung, Y.-N., Chen, J.-S., Lin, Y.-H., Huang, R.-Y., Lin, C.-Y., Ko, N.-Y. (2022). Psychological distress and protective behaviors during the COVID-19 pandemic among different populations: Hong Kong general population, Taiwan healthcare workers, and Taiwan outpatients. Frontiers in Medicine, 9, 800962.

Alijanzadeh, M., Ahorsu, D. K., Mahmoudi, N., Majd, N. R., Alimoradi, Z., Griffiths, M. D., Lin, C.-Y., Liu, H.-K., & Pakpour, A. H. (2021). Fear of COVID-19 and trust in the healthcare system mediates the association between individual’s risk perception and preventive COVID-19 behaviours among Iranians. International Journal of Environmental Research and Public Health, 18, 12146.

Pakpour, A. H., Liu, C.-h., Hou, W.-L., Chen, Y.-P., Li, Y.-P., Kuo, Y.-J., Lin, C.-Y., Scarf, D. (2021). Comparing fear of COVID-19 and preventive COVID-19 infection behaviors between Iranian and Taiwanese older people: Early reaction may be a key. Frontiers in Public Health, 9, 740333.

Chang, K.-C., Strong, C., Pakpour, A. H., Griffiths, M. D., & Lin, C.-Y. (2020). Factors related to preventive COVID-19 infection behaviors among people with mental illness. Journal of the Formosan Medical Association, 119(12), 1772-1780.

Lin, C.-Y., Imani, V., Majd, N. R., Ghasemi, Z., Griffiths, M. D., Hamilton, K., Hagger, M. S., & Pakpour, A. H. (2020). Using an Integrated Social Cognition Model to Predict COVID-19 Preventive Behaviours. British Journal of Health Psychology, 25(4), 981-1005.

RESPONSE: Based on your recommendations and our own readings, we have added new text and 19 new citations to the Introduction, including four of the suggested citations above. Please see lines 106-139 for these revisions. We hope our revised text provides a more thorough overview of past research on protective behavior, including recent studies of COVID-19 behavior.

2. I think that the authors may want to discuss the issue of behavioral fatigue in the preventive behaviors of COVID-19.

Harvey N. (2020). Behavioral Fatigue: Real Phenomenon, Naïve Construct, or Policy Contrivance?. Frontiers in psychology, 11, 589892.

RESPONSE: Thank you, we agree behavioral fatigue deserves some discussion in this paper. We now acknowledge the issue in our text on study limitations (see lines 525-527) and in our text on future research directions (see lines 544-548). 

3. I wonder if the authors can provide information regarding whether the measures are reliable and valid.

RESPONSE: To address this concern, we have made a number revisions to the text, detailed below:

• Lines 199-201: We added a description of the survey team’s substantive and methodological expertise.

• Lines 222-228: We added information and citations that speak to the validity and reliability of our outcome (social-distancing and masking) measures.

• Lines 290-291: We note, with citations, that our informational trust questions mirror those used in prior studies.

• Lines 311-317: We added information and citations that speak to the validity and reliability of our health belief measures

• Lines 323-330: We now explicitly note where survey measures of interpersonal and community context draw on validated scales.

4. I wonder why the authors used cluster analysis instead of latent class analysis to classify the participants into different groups. I believe that using latent class analysis can tackle the issue of repeated measures and provide a more accurate classification.

RESPONSE: As you point out, latent class analysis is an alternative way to cluster survey respondents and investigate relationships between respondent characteristics and cluster membership. In future work, we plan to cluster respondents by the longitudinal trajectories of their response to these survey items, and for this purpose latent class analysis would be especially useful. In this paper we were focused on the somewhat simpler problem of clustering on the basis of time-invariant (i.e., averaged) responses, for which we felt hierarchical clustering was a simpler but sufficiently flexible approach.

5. Can the authors provide some information regarding the sample representativeness?

RESPONSE: Per your recommendation, we have added a more detailed description of the UAS sampling and recruitment approach to the Methods section. See lines 180-188. 

To address sample representativeness, we have also made the following revisions:

(1) We added a column with total sample demographic characteristics to Table 1.

(2) We moved the description of the analytic sample from the Results section to the Methods section (see lines 207-220). 

(3) We added a table with select sample characteristics, presented alongside population benchmarks from the 2020 ACS 5-Year Estimates, to the S1 Appendix. 

6. Again in the Discussion, I feel that the authors did not compare their results with the existing evidence on the protective behaviors during COVID-19 pandemic.

RESPONSE: Thank you, we have made a number of revisions to the Discussion to more thoroughly compare our results to existing evidence on protective behavior. See lines 455-463 and lines 480-510 for these revisions. 

REVIEWER #2 (copied/pasted from comments provided in manuscript)

Overall, the paper is interesting but it needs further improvement and clarification as to how this study is needed (contribution to the existing knowledge is unclear due to poor explanation of research gaps).

1. Lines 124-130 on research gaps: please rejustify the gaps as using SEM in covid has been Done. What is a new thing can be offered by this study?

“First, the majority of studies examine protective behavior at a single point in time, or they assess the predictors or protective-behavioral change between time points.”

a. Explain how the study fills this gap

“Second, most studies take a variable-centered, versus person-centered, approach to measuring protective behavior. They assume behaviors like masking and social distancing relate to one another in a uniform way across the population”

a. need more explanations. At the end of the day, researchers will still use variables or factors to determine the compliant behaviour. I would think this is more of an approach or theory used to study the matters. (phenonmenological approac?)

RESPONSE: Thank you, we agree that our research contributions were not well-articulated in the original manuscript and we have made a number of revisions to address this. Please see lines 153-174. We hope these changes better clarify the two gaps we see in the research literature and better explicate how our study fills those gaps.

Please note that in the process of revising this part of the paper, we decided to remove the text on “variable-centered” versus “person-centered” approaches. Upon review, we felt this text confused more than clarified and was not worth highlighting.

3. Outcome measures: tabulate the questions and scale of the data

RESPONSE: To address this comment, we have made a number revisions to the text, detailed below:

• Lines 230-238: We added response options and question means and SD’s for each of the social-distancing behavior items.

• Lines 245-249: We added questions means and SD’s for each of the masking behavior items. 

We also added response scales where they were previously missing:

• Trusted information sources: Lines 288-289

• Health beliefs: Lines 298-310

4. Discussion: discussion needs to be improved as to how the above key findings are compared... summary of the key findings can be included (meaning....which predictors are ultimately included as opposed to the conceptual framework).

RESPONSE: Thank you for this recommendation. We have made a series of revisions to provide readers with a more detailed summary of our key findings. These revisions are summarized below:

(1) Lines 406-418: We added more text to our Results section to clarify which predictors which were not statistically significant in our regression models.

(2) Lines 452-479: We added a more thorough summary of our key findings to the Discussion. 

(3) Lines 455-463, 480-510: We added text to the Discussion to more thoroughly compare our results to existing evidence on protective behavior. 

5. Conclusion: please include limitations of this study and suggest future studies

RESPONSE: In addition to the text on limitations in the Discussion, we have added a brief outline of limitations to the Conclusion (see lines 536-538). We have also added future research directions to the Conclusion (see lines 538-562).

6. Need to improve the conceptual framework figure by including the dependent variable (adherence) into the framework.

RESPONSE: Thank you, we appreciate this suggestion. We have revised Figure 1 to include our dependent variable, adherence to protective behavior.

---

## [Decision Letter · Decision Letter 1]

8 Jun 2023

Behind Closed Doors: Protective Social Behavior During the COVID-19 Pandemic

PONE-D-22-23931R1

Dear Dr. Thomas,

We’re pleased to inform you that your manuscript has been judged scientifically suitable for publication and will be formally accepted for publication once it meets all outstanding technical requirements.

Kind regards,

Celia Andreu-Sánchez

Academic Editor

PLOS ONE

Additional Editor Comments (optional):

Reviewers' comments:

Reviewer's Responses to Questions

**Comments to the Author**

1. If the authors have adequately addressed your comments raised in a previous round of review and you feel that this manuscript is now acceptable for publication, you may indicate that here to bypass the “Comments to the Author” section, enter your conflict of interest statement in the “Confidential to Editor” section, and submit your "Accept" recommendation.

Reviewer #1: All comments have been addressed

Reviewer #2: All comments have been addressed

2. Is the manuscript technically sound, and do the data support the conclusions?

Reviewer #1: Yes

Reviewer #2: Yes

3. Has the statistical analysis been performed appropriately and rigorously? 

Reviewer #1: Yes

Reviewer #2: Yes

4. Have the authors made all data underlying the findings in their manuscript fully available?

Reviewer #1: Yes

Reviewer #2: Yes

5. Is the manuscript presented in an intelligible fashion and written in standard English?

Reviewer #1: Yes

Reviewer #2: Yes

6. Review Comments to the Author

Reviewer #1: The authors have satisfactorily addressed all my prior concerns. The revised manuscript is acceptable in my opinion.

Reviewer #2: I believed the paper has been revised according to the reviewer's comments. Therefore i would suggest the paper to be accepted

7. PLOS authors have the option to publish the peer review history of their article (what does this mean?). If published, this will include your full peer review and any attached files.

Reviewer #1: No

Reviewer #2: No

---

## [Editor Report · Acceptance letter]

16 Jun 2023

PONE-D-22-23931R1 

Behind Closed Doors: Protective Social Behavior during the COVID-19 Pandemic 

Dear Dr. Thomas:

I'm pleased to inform you that your manuscript has been deemed suitable for publication in PLOS ONE. Congratulations! Your manuscript is now with our production department. 

Kind regards, 

on behalf of

Dr. Celia Andreu-Sánchez 

Academic Editor

PLOS ONE